# Survey of Ticks and Tick-Borne Rickettsial and Protozoan Pathogens in Eswatini

**DOI:** 10.3390/pathogens10081043

**Published:** 2021-08-17

**Authors:** Kimberly J. Ledger, Lorenza Beati, Samantha M. Wisely

**Affiliations:** 1Department of Wildlife Ecology and Conservation, University of Florida, Gainesville, FL 32611, USA; kimberly.ledger@ufl.edu; 2US National Tick Collection, Institute for Coastal Plain Science, Georgia Southern University, Statesboro, GA 30458, USA; lorenzabeati@georgiasouthern.edu

**Keywords:** ticks, *Rickettsia*, *Anaplasma*, *Ehrlichia*, *Babesia*, *Hepatozoon*, *Theileria*, land use, Eswatini

## Abstract

Ticks are widespread parasites of vertebrates and major vectors of pathogens to humans, domestic animals, and wildlife. In southern Africa, numerous tick species transmit diseases of economic and health importance. This study aimed to describe the occurrence of ticks and tick-borne pathogens in multiple land-use types and the possible role of ticks in the transmission of pathogen species. Using molecular techniques, we screened 1716 ticks for infection by rickettsial bacteria and protozoans. To characterize pathogen identity, we sequenced multiple loci from positive samples and analyzed sequences within a phylogenetic framework. Across the seven tick species collected as nymphs or adults, we detected *Rickettsia*, *Anaplasma*, *Ehrlichia*, *Babesia*, *Hepatozoon*, and *Theileira* species. We found that some tick species and tick-borne pathogens differed according to land use. For example, we found a higher density of *Haemaphysalis elliptica* and higher prevalence of *Rickettsia* in *H. elliptica* collected from savanna grasses used for livestock grazing near human settlements than savanna grasses in conservation areas. These findings highlight the importance of comprehensive surveillance to achieve a full understanding of the diversity and ecology of the tick-borne pathogens that can infect humans, domestic animals, and wildlife.

## 1. Introduction

Vector-borne diseases are among the most important risks to animal and human health worldwide. Hard ticks (Acari:Ixodidae Koch, 1844) are efficient vectors of a variety of pathogens, including bacteria (e.g., *Rickettsia* da Rocha-Lima, 1916, *Anaplasma* Theiler, 1910), viruses (e.g., *Flaviviruses*, *Nairovirus*), and protozoa (e.g., *Babesia* Starcovici, 1893, *Theileria* Bettencourt, França & Borges, 1907), which cause major diseases affecting humans, livestock, wildlife, and companion animals [1]. In livestock, diseases transmitted by ticks cause major constraints on and financial problems for animal production, particularly in developing countries in tropical and sub-tropical regions of the world [2]. In humans, ticks have great impacts on public health. Insufficient surveillance for ticks and tick-borne diseases in many regions of the world can result in underdiagnosis and gaps in the data on tick-borne infections. Surveillance of ticks and their pathogens is a crucial step towards effective investigation and management of tick-borne diseases [3].

Land use is linked to ticks and tick-borne diseases through habitat availability and quality for both the vertebrate host and the tick [4,5]. Host habitat preferences influence where ticks occur; and when ticks are not on a host, they require adequate vegetation for temperature regulation and water balance for survival in the environment [6]. These microclimate conditions also influence tick-borne pathogen transmission [7,8]. Rapid and substantial land-use change has occurred throughout southern Africa as urbanization and modern agricultural production techniques have spread throughout the region [9,10]. This patchwork of habitat types, including commercial cattle ranches, monoculture agriculture, and conservation areas, are interspersed with traditional human settlements and grazing lands which creates divergent communities of large vertebrate hosts available to ticks [11]. Tick species do not feed equally on all vertebrate animals; therefore, their distribution should reflect the composition of available hosts. In the Lowveld region of Eswatini in southern Africa, numerous species of questing ticks occur in savannas and the landscape influences their occupancy and abundance in species-specific ways [12].

Ticks are the main vector for numerous bacteria from the order Rickettsiales Gieszczykiewicz, 1939 and infections from *Rickettsia*, *Anaplasma*, and *Ehrlichia* Moshkovski, 1947 are widely reported in southern Africa. Most tick-borne pathogen species that circulate between ticks and vertebrates have been reported in the livestock, domestic, or wild animal hosts they infect but surveillance of these pathogens in tick populations occurs less often [13,14,15,16,17,18,19,20,21,22]. More recently, emphasis has been placed on pathogen detection in the tick itself [23,24,25,26,27,28,29,30]. Though designation of a tick species as a true vector requires transmission experiments, pathogen detection from host-seeking field-collected ticks is a first step in identifying candidate vector species.

In southern Africa, there are four tick-borne spotted fever group (SFG) *Rickettsia* species associated with human disease: *Rickettsia africae* Kelly et al. 1996 (African tick bite fever (ATBF)), *Rickettsia conorii* Brumpt, 1932 (Mediterranean spotted fever), *Rickettsia aeschlimannii* Beati et al. 1997 (innominate rickettsioses), and *Rickettsia sibirica subsp.*
*mongolotimonae* Zdrodovskii, 1948 (lymphangitis-associated rickettsioses) [31,32,33,34,35,36]. ATBF is the leading cause of fever among travelers to South Africa [37] and numerous cases have been documented in travelers who visited Eswatini [38,39,40,41,42]. Reports of ATBF in local African populations are scarce [37], but seroprevalence of past *Rickettsia* infection from cattle herders and acute febrile illness patients is high (92.2% and 63.4%, respectively) [43], suggesting significant risk in rural settings where the primary vector, *Amblyomma hebraeum* Koch, 1844, is commonly detected. SFG *Rickettsia* are relatively unique in their efficient transovarial transmission (passage from the adult female through the ovaries to the unfed larvae of the next generation), as infected females of some tick species give rise to at least one positive egg 100% of the time [44,45,46]. Therefore, SFG *Rickettsia* are less dependent on vertebrate hosts as pathogen reservoirs and the tick itself can be considered a reservoir.

Numerous tick-borne pathogens belonging to the family Anaplasmataceae are of major concern due to their importance to veterinary medicine in Africa. Well-known livestock diseases belonging to this family include bovine, ovine, and caprine anaplasmosis and heartwater. Reports of newly identified and unclassified *Anaplasma* and *Ehrlichia* species in southern Africa are common [47,48,49,50,51]. In general, Anaplasmataceae are intracellular bacteria that circulate in the blood of wild and domestic animals and are acquired by the tick via horizontal transmission during blood feeding.

Ticks can also vector intracellular apicomplexan parasites. Apicomplexan tick-borne diseases are caused by *Babesia*, *Theileria*, *Hepatozoon* Miller, 1908, *Hemolivia* Petit et al. 1990, and *Cytauxzoon felis* Kier, 1979 [52,53]. Of these genera, *Theileria* and *Babesia* cause some of the most economically important diseases in domestic and wild animals [54]. Ticks primarily acquire *Theileria* by horizontal transmission and *Babesia* via either transovarial or horizontal transmission. In southern Africa, numerous *Babesia* species are known to cause babesiosis in animals and are emerging zoonoses in humans. *Theileria* species cause theileriosis in a range of domestic and wild ungulates [54]. Numerous non-pathogenic *Theileria* species are considered benign, though there is not always consistency within and/or among mammalian host species. *Hepatozoon* species are widely reported from amphibians, reptiles, birds, and mammals; in particular, from wild carnivores [55].

Ticks and tick-borne diseases are linked to the presence of their vertebrate hosts and a suitable microclimate in the environment, both of which are sensitive to landscape changes [56]. This study reports the occurrence of questing ticks and their bacterial and protozoan pathogens in the Lowveld region of Eswatini in southern Africa. We sampled ticks from savanna grasses in protected areas, including conservation areas and mixed cattle and game ranches, and unprotected areas, including cattle-only ranches and communal rangelands. The protected area sites are characterized by the presence of large-bodied wildlife and limited human activity while the unprotected area sites are characterized by the presence of livestock, domestic animals, and humans. The purpose here is to describe the occurrence of ticks and tick-borne pathogens and the possible role of ticks in the transmission of pathogen species. Surveillance of ticks and their pathogens in the context of land use aims to inform how land-use practices influence disease risk for humans, livestock, and wildlife.

## 2. Results

### 2.1. Tick Diversity

A total of 1009 adults, 3268 nymphs, and approximately 81,000 larvae were collected during this study (Appendix A). Adult ticks were identified morphologically to species-level and included *Amblyomma hebraeum*, *Haemaphysalis elliptica* Koch, 1844, *Rhipicephalus appendiculatus* Neumann, 1901, *Rhipicephalus evertsi* Neumann, 1897, *Rhipicephalus maculatus* Neumann, 1901, and *Rhipicephalus simus* Koch, 1844. Nymph and larval ticks were morphologically identified to genus-level and included *Amblyomma* and *Rhipicephalus* Letreille, 1806. Molecular analysis of tick genes confirmed the morphological identification of all species, as well as revealing the presence of *Rhipicephalus muehlensi* Zumpt, 1943 (Figure 1, Table 1). All extracted *Amblyomma* larvae were identified as *A. hebraeum*. There was 99–100% sequence identity for all species in this study to sequences from Genbank, except for *H. elliptica*, which had 96–98% sequence identity to the limited number of publicly available reference sequences.

Using a restriction fragment length polymorphism (RFLP) assay, we identified 39 adult *R. muehlensi* from a portion of the adult ticks morphologically identified as *R. appendicualtus* and 554 *R. appendiculatus*, 41 *R. maculatus*, and 74 *R. muehlensi* nymphs from the extracted subset of *Rhipicephalus* nymphs (n = 669). We identified four *Rhipicephalus microplus* (Canestrini, 1888) and 34 *Rhipicephalus decoloratus* Koch, 1844 from the extracted subset of *Rhipicephalus* (*Boophilus*) larvae (n = 38) and 28 *R. appendiculatus*, 2 *R. maculatus*, and 9 *R. muehlensi* from the extracted subset of *Rhipicephalus* larvae (n = 39) (Figure 2). 

Tick species richness was greatest in wildlife conservation areas, with seven species collected as adults and/or nymphs, as well as *R. decoloratus* from the subset of identified *Rhipicephalus* (*Boophilius*) larvae. We recorded five species collected as adults and/or nymphs in mixed cattle and game ranches, as well as *R. decoloratus* from the subset of identified *Rhipicephalus* (*Boophilius*) larvae. We recorded four species collected as adults and/or nymphs in cattle-only ranches and communal areas, as well as *R. microplus* from communal areas from the subset of identified *Rhipicephalus* (*Boophilius*) larvae.

The density of adult *H. elliptica* was significantly greater in unprotected versus protected sites (t(25) = −3.6, *p* = 0.001). The densities of *Rhipicephalus* nymphs and *Rhipicephalus* larvae (including *R. appendiculatus*, *R. maculatus*, and *R. muehlensi*) were greater in protected versus unprotected sites ([*Rhipicephalus* nymphs: t(25) = 2.6, *p* = 0.01; *Rhipicephalus* larvae: t(25) = 4.0, *p* < 0.001). The densities of adult *R. appendiculatus* and adult *R. simus* were greater in protected sites, but the relationship was not significant (*R. appendiculatus* adults: t(25) = 1.2, *p* = 0.25; *R. simus* adults: t(25) = 0.7, *p* = 0.5). The density of *Amblyomma* larvae and *Rhipicephalus* (*Boophilus*) larvae were greater in unprotected sites, but the difference was not statistically significant (*Amblyomma* larvae: t(25) = −1.1, *p* = 0.3; *Rhipicephalus* (*Boophilus*) larvae: t(25) = −0.2, *p* = 0.8) (Appendix A). *Amblyomma hebraeum* nymphs and adults were not included in this analysis as they are host-seeking ticks and collection from vegetation by dragging would likely not be representative of true counts. *Rhipicephalus evertsi*, *R. maculatus*, and *R. muehlensi* were only collected from protected sites.

The density of *Rhipicephalus* (*Boophilus*) larvae was significantly greater in summer 2018 than winter 2018 (t(10) = −4.9, *p* < 0.001) and in winter 2019 than winter 2018 (t(10) = −3.1, *p* = 0.01). The density of *Rhipicephalus* larvae was significantly greater in winter 2018 than summer 2018 (t(10) = 2.3, *p* = 0.05). The density of *Rhipicephalus maculatus* adults was significantly greater in summer 2018 than winter 2018 (t(10) = −3.1, *p* = 0.01) and in winter 2018 than winter 2019 (t(10) = 2.6, *p* = 0.03). The densities of all other tick species and life stages lacked statistically significant variation by season or year (Appendix A).

### 2.2. Tick-Borne Pathogen Diversity

We screened 1716 individuals (adults = 1009, nymphs = 707) for bacterial and protozoan pathogens and detected *Anaplasma*, *Ehrlichia*, *Rickettsia*, *Babesia*, *Heptatozoon*, and *Theileria* pathogen DNA. We detected three described species of *Rickettsia*, one candidate *Rickettsia* species, and three undescribed genotypes of *Rickettsia* in *A. hebraeum*, *H. elliptica*, and *R. simus* (Figure 3, Table 2, and Appendix A). Sequencing of the Anaplasmataceae PCR products determined the presence of two described species and 13 unique genotypes of undescribed Anaplasmataceae in *Amblyomma hebraeum*, *H. elliptica*, *R. appendiculatus*, *R. maculatus*, *R. simus*, and *R. muehlensi* (Figure 4, Table 3, and Appendix A). Sequencing of the Apicomplexa PCR products determined the presence of 14 different known species or genotypes and four undescribed genotypes of protozoan parasites in the genera *Babesia*, *Heptatozoon*, and *Theileria*. *Rhipicephalus appendiculatus* and *R. simus* tested positive for *Babesia* sp., *R. simus* and *R. muehlensi* tested positive for *Hepatozoon* sp., and *A. hebraeum*, *R. appendiculatus*, *R. maculatus*, *R. simus*, and *R. muehlensi* tested positive for *Theileria* sp. (Figure 5, Table 4, and Appendix A). 

We identified the greatest richness of pathogen strains from *R. appendiculatus* (seven Anaplasmataceae and nine Apicomplexa) and *R. simus* (four *Rickettsia*, four Anaplasmataceae, and seven Apicomplexa). The highest prevalence estimate was of *R. africae* in *A. hebraeum* nymphs (19/38 = 50%).

We detected *Rickettsia*, *Anaplasma*, and *Ehrlichia* at sites in both protected and unprotected areas, while detection of any Apicomplexa (*Babesia*, *Hepatozoon*, and *Theileria*) was limited to protected areas only. The only pathogen to be exclusively found in unprotected sites was *R. conorii*. A larger proportion of *H. elliptica* were infected with *R. conorii* or *R. massiliae* in unprotected sites than in protected sites (Chi-sq = 14.3, *p* < 0.001). A larger proportion of *H. elliptica* were also infected with *Anaplasma* in unprotected sites versus protected sites but this relationship was only marginally significant (Chi-sq = 3.5, *p* = 0.06). No other comparisons of tick infection status and land-use classification were significant.

The proportion of infected ticks (number of infected ticks/number of tested ticks) varied from 0% (0/28) to 60% (3/5) per site. Of the resampled sites, the proportion of infected ticks was significantly greater in winter 2018 than summer 2018 (t(10)= −5.5; *p* < 0.001), but there was no significant difference between winter 2018 and winter 2019 (t(10) = 1.7, *p* = 0.1). Of the sites sampled in winter 2018, the overall proportion of infected ticks from protected sites was 17.7% and from unprotected sites it was 20.6% (t(10) = −0.6; *p* = 0.6) (Appendix A).

Co-infection of two tick-borne pathogens within an individual tick was observed in 1.2% (21/1716) of the ticks tested. Co-infections of *Anaplasma* + *Rickettsia* were 0.41% (7/1716), *Ehrlichia* + *Rickettsia* were 0.06% (1/1716), *Theileria* + *Rickettsia* were 0.29% (5/1716), *Anaplasma* + *Theileria* were 0.06% (1/1716), and *Ehrlichia* + *Theileria* were 0.41% (7/1716) (Appendix A).

## 3. Discussion

In this study we report the occurrence of questing tick species from wildlife conservation areas, mixed cattle and game ranches, cattle-only ranches, and communal lands in the Lowveld region of Eswatini. Furthermore, we detected thirty-nine unique tick-borne pathogen species or genotypes, as well as identified numerous novel associations between bacterial and protozoan pathogens and tick vectors using molecular detection methods.

This study updates tick occurrence data in a region with limited published records and describes the widespread occurrence of several generalist ticks (i.e., *A. hebraeum*, *H. elliptica*, *R. appendiculatus*, and *R. simus*) and the presence of less studied tick species associated with wildlife conservation areas (i.e., *R. maculatus* and *R. muehlensi*). Overall, we collected nine of the fourteen species of hard ticks that have published records from Eswatini [57,58,59,60,61]. Molecular analysis of two mitochondrial genes (*12S* and *CO1*) and one nuclear gene (*ITS2*) confirmed our morphological tick identifications and further differentiated difficult-to-identify species and life stages of *Rhipicephalus* ticks. Traditionally, tick identification has been based on morphological characters, which can be challenging when recently diverging tick species have intraspecific polymorphisms [62]. We demonstrated that the PCR-RFLP diagnostic originally described by Mtambo et al. [63] can differentiate numerous species of *Rhipicephalus* and may be particularly useful in identifying immature life stages. Accurate and meaningful surveillance to identify ticks as vectors for infectious diseases in humans, livestock, and wildlife requires precise identification of ticks to inform descriptions of their distribution and ecology. This study utilized phylogenetic analyses to visualize systematic relationships between our sequences and homologous reference sequences. Phylogenetic relationships among our *Rhipicephalus* ticks were generally well-supported and consistent with previous phylogenetic studies [64,65]. Molecular sequences of *R. muehlensi* are limited and, until recently, the species was not included in molecular phylogenic trees [65]. Our molecular grouping of *R. muehlensi* with the *R. appendiculatus* group is consistent with morphological grouping [62]. This study also contributes important genetic resources for tick species with limited publicly available reference sequences, including the first published sequence of *CO1* for *H. elliptica* and *ITS2* for *R. muehlensi*.

Ticks were collected from habitats with distinct communities of vertebrate hosts, which appear to have also shaped the tick communities within each habitat type. In this study, larval *A. hebraeum*, which are considered generalist ticks that feed on both domestic and wild animals, occurred in all habitat types. *Haemaphysalis elliptica*, a specialist parasite of carnivores, was found at higher densities in unprotected savannas than in protected savannas, likely due to the large number of domestic dogs associated with cattle ranches and human habitation [66]. *Haemaphysalis elliptica* have been collected in nearly every survey of ticks on domestic dogs in South Africa [67] and were the predominant species among ticks collected from dogs in peri-urban areas that have gardens and farms where rodents (an important host of immature stages of *H. elliptica*) thrive [62,68]. We collected *R. appendiculatus* (generalist adults and immatures) and *R. simus* (generalist adults) [67] from conservation and communal areas and conservation, communal, and cattle ranches, respectively. *Rhipicephalus* nymphs, consisting of *R. appendicualtus*, *R. maculatus*, and *R. muehlensi*, were significantly more abundant in protected areas, possibly due to higher densities of hosts, the lack of acaracide treatments in conservation areas, and the association of *R. maculatus* and *R. muehlensi* with wildlife hosts. Finally, adult *R. maculatus* and *R. muehlensi*, which feed exclusively on wildlife, were only found in protected areas where large-bodied wildlife was present. *Rhipicephalus microplus* is a specialist parasite of domestic cattle and goats and our only records were larvae from communal lands. *Rhipicephalus decoloratus* is a feeding generalist and regularly infests wild and domestic ungulates, carnivores, rodents, and birds, which was reflected our identification of larvae from conservation areas and cattle ranches.

This study documented pathogens known to be circulating in southern Africa, a diversity of undescribed bacterial and protozoan genotypes, and novel records of tick-pathogen associations. Rickettsioses, such as ATBF, are commonly reported in international travelers and are an underdiagnosed burden on the health of local populations. The widespread occurrence of *A. hebraeum* ticks and high prevalence of *R. africae* (up to 50%) in this study corroborate the ongoing public health concern. The high prevalence of *R. africae* in *Amblyomma* ticks is due to highly efficient transovarial transmission. Additionally, *A. hebraeum* larval ticks are the principal vector of ATBF in southern Africa because they are collected from humans 10× more than any other tick species at any life stage [69]. We also documented *R. conorii*, the agent of Mediterranean spotted fever, and *R. massiliae*, recently documented to cause a mild spotted fever rickettsioses [70,71], in *H. elliptica*. To our knowledge, this observation is the first record of *R. massiliae* in *H. elliptica*. We also documented three unique genotypes of undescribed *Rickettsia* with unknown pathogenicity. These *Rickettsia* genotypes may represent a potential pathogen or a non-pathogenic endosymbiotic rickettsiae.

*Rhipicephalus simus* was the only species of *Rhipicephalus* from which we found *Rickettsia*. We detected *R. massiliae* and Candidatus *Rickettsia barbariae*, which have both been previously described in *R. simus* [26,72], as well as two undescribed genotypes of *Rickettsia* sp. Though a known vector, we did not detect *Rickettsia conorii* in any of our *R. simus* [73]. *Haemaphysalis elliptica* and *R. simus* adults are among the most common ticks to attach to and feed on humans in South Africa [69], further raising concern for transmission of pathogenic *Rickettsia* in humans. To date, the only described human cases of rickettsioses caused by *R. massiliae* come from France and Argentina, but the pathogen has been detected in *Rhipicephalus sanguineus* Latreille, 1806 from Morocco [74] and the Ivory Coast [75]. The recent report of *R. massiliae* in *Amblyomma sylvaticum* De Geer, 1778 and *R. simus* [26], and now *H. elliptica* in southern Africa, suggest that more studies are needed to investigate the distribution and vectors of this pathogen. Overall, SFG *Rickettsia* are underreported and underappreciated causes of illness in local populations in southern Africa.

*Anaplasma* and *Ehrlichia* infections in mammalian hosts may range from unapparent infection to severe disease and mortality. Our detection of *E. ruminantium*, the agent of heartwater, in an *A. hebraeum* nymph is of veterinary health importance as heartwater is one of the major causes of stock loss in Sub-Saharan Africa. In addition to *E. ruminantium*, *E. minasensis* is the only other species of *Ehrlichia* known to naturally infect cattle and cause ehrlichiosis [76,77]. *Ehrlichia minasensis* has previously been described in *R. appendiculatus* ticks from South Africa [29], but this is the first known record in both *R. muehlensi* and *R. simus*. *Ehrlichia minasensis* is known to also infect cervids [78] and dogs [79]. We identified numerous positive samples for *Anaplasma* sp. and *Ehrlichia* sp. using the *16S*, *rpoB*, and/or *groEL* genes. However, phylogenetic comparison of the sequences to known *Anaplasma* species was unable to resolve the identity of the samples, indicating the possibility of substantial undescribed diversity of *Anaplasma*-like and *Ehrlichia*-like species in ticks.

*Theileria* is a genus of tick-transmitted parasitic protozoa that includes highly pathogenic to non-pathogenic species. Our study did not detect any highly pathogenic species and detected only one species that is occasionally pathogenic to cattle, *T. taurotragi* from *R. appendiculatus* and *R. muehlensi*. To our knowledge, this is the first report of *T. taurotragi* in *R. muehlensi*. *Theileria taurotragi* seems to be cattle- and eland-specific, although it has been detected in bushbuck, *Tragelaphus sylvaticus* (Pallas, 1766), in Uganda [80]. While there are no eland in our study area of Eswatini, our detection of *T. taurotragi* from multiple wildlife conservation areas may be associated with bushbuck or cattle that occasionally utilize the land.

We detected numerous non-pathogenic (to cattle) *Theileria* species and genotypes in this study. Of the previously described pathogen–vector associations, we detected *T. velifera* in *A. hebraeum* ticks. Cattle and African buffalo are the only described vertebrate hosts of *Theileria velifera* [21,81]. Our detection of *T. velifera* from a conservation area and a mixed cattle and game ranch, neither of which have African buffalo, suggests that cattle that occasionally cross into conservation areas, or another vertebrate host, are involved in pathogen maintenance. We detected *Theileria* sp. (sable), which is a benign species to cattle but causes mortality in naïve roan antelope and sable antelope. *Theileria* sp (sable) has been detected in numerous vertebrate hosts including dog, cattle, African buffalo, blue wildebeest, and nyala [14,15,18,20,82]. All Antilopinae Gray, 1821, a subfamily of Bovidae, may be potential hosts of *T*. sp (sable). *R. appendiculatus* has been implicated as a vector of *T*. sp (sable) [83], but this is the first known report of *T*. sp (sable) from *R. simus*. *Theileria ovis* is a benign *Theileria* species found in sheep, goats, cattle, and dogs [25,84,85]. The vectors and any potential wildlife hosts have not yet been identified [86]. We detected *T. ovis* in *R. appendiculaus* nymphs from multiple conservation areas. We detected *Theileria* sp (waterbuck), a piroplasmid first identified in waterbuck from Kenya, from *A. hebraeum* in multiple conservation areas. Waterbuck are known to be present in the study area and recent work suggests that *T. ovis*, *T.* sp (sable), and *T.* sp (waterbuck) may not be host-specific within the Antilopinae [87].

*Theileria equi*, a causative agent of equine piroplasmosis in horses, mules, donkeys, and zebras, is a reportable disease of international importance that negatively affects the movement of horses and other equines for trade and sporting. Currently, there are at least 25 suspected or confirmed tick vectors of *T. equi* belonging to six different genera including *Rhipicephalus* [88], but this study reports the first detection in both *R. appendiculatus* and *R. muehlensi*. Zebras are the only equine present in conservation areas within the study area.

We detected *Theileria* sp (giraffe), a piroplasmid found in southern African giraffe [16], in *R. appendiculatus.* To our knowledge, this is the first description of one of the many *Theileria* genotypes associated with giraffes to be documented from *R. appendiculatus*. We also report the first detections of *Theileria* sp Tragelaphini (B, D, and F), which are exclusive to Tragelaphini; *Theileria* sp Aepycerotini, which is specific to impala [87], in *R. appendiculatus*, *R. maculatus*, *R. simus*, and *R. muehlensi*; and *Theileria* sp. (impala), another impala-specific genotype, in *A. hebraeum*. Finally, our detection of *Theileria* sp (rodent) from a *R. simus* is notable because, while rodents are recognized as reservoirs of numerous infectious pathogens, they are not commonly thought of as reservoirs for *Theileria*. However, there have been recent reports of *Theileria* in small mammals [89,90,91] and immature stages of *R. simus* feed on small mammals [92].

Ticks were identified as vectors of *Babesia* species over a century ago [93], but identification of the particular tick species associated with both well-known and novel species and genotypes remains patchy [94]. Two of our *Babesia* sp. genotypes, one from three *R. simus* and the other from two *R. appendiculatus*, are most similar to *Babesia caballi*, an equine piroplasm, known to infect horses in South Africa [95]. Numerous tick species are known to transmit *B. caballi*, including species of *Rhipicephalus* ticks, but this is the first known report of a *Babesia* sp. similar to *B. caballi* in *R. appendiculatus* or *R. simus*. The third *Babesia* sp. genotype, in this study, detected in a *R. simus*, is most similar to *Babesia* sp. Suis, originally isolated from a pig [96]. This *Babesia* genotype has also been documented in South Africa as the cause of fatal porcine babesiosis in a pot-belled pig [97]. *Rhipicephalus sanguineus* and *Rhipicephalus bursa* have been implicated as potential vectors for *Babesia* sp. Suis [96], but this is the first reported association with *R. simus*. *Rhipicpehalus simus* is a known vector of *Babesia trautmanni*, another cause of porcine babesiosis [98].

*Hepatozoon* species are distributed worldwide and reported in numerous wild canids and felids [99,100,101], rodents [102,103,104], and reptiles [55,105,106,107], including reports from caracals [108], African wild dogs [109], black-backed jackals (*Canis mesomelas*) [17], a gerbil (*Gerbilliscus leucogaster*) [110], and frogs [111] in southern Africa. Many species of ticks are considered main or potential vectors of *Hepatozoon* species, including members of the genera *Rhipicephalus*, *Haemaphysalis*, and *Amblyomma* [112,113]. Our detection of *Hepatozoon* sp. is at least the second in *R. simus* [114] and the first known detection of *Hepatozoon* sp. in *R. muehlensi*.

Seasonal patterns in tick abundances have been described in southern Africa for over half a century [115]. While seasonal trends exist in tick abundances, they are specific to a tick species and the location of sampling and often differ between collections from domestic or wild animals or from the vegetation [67]. We also documented year-to-year variability in the density of *Rhipicephalus* (*Boophilus*) larvae and *R. maculatus* adults in our study. These differences may be explained by the on average 1 °C warmer temperatures and 25% less rain in Eswatini during the winter of 2019 than the winter of 2018 [116], as *Rhipicephalus* (*Boophilus*) are known to be active in warm temperatures and *R. maculatus* are associated with cool, moist climates [67]. Overall, the changes in tick numbers over time likely reflect differences in both their host communities and in climatic conditions of hosts and ticks. Our comparison of tick density amongst sites in different land uses was limited to the winter season of 2018. Despite this temporal limitation, our study does provide comparative data on tick density across sites at one point in time as seasonal variation is not expected to differ by land-use type.

We utilized phylogenetic analyses to visualize systematic relationships between our sequences and homologous reference sequences for pathogen identification. The low (<70%) bootstrap values at some nodes of the pathogen phylogenetic trees indicate the genes used in this study may not provide sufficient information for accurate reconstruction of evolutionary relationships. The lack of support may be due to recent divergence and incomplete separation of clades at the marker gene.

Wildlife areas could act as the source of certain ticks and pathogens to domestic animals in the surrounding areas. Wild animals can amplify tick populations and be the source of pathogen infection in ticks [117]. For example, wild ungulates, some of which are carriers of *Theileria* species, are the preferred hosts of *R. appendiculatus* and *R. simus*, but when given the opportunity these ticks will parasitize cattle and can transmit disease causing *Theileria* species [62]. This study detected numerous pathogen genotypes in ticks collected from protected area indicating a risk of pathogen spread to naïve hosts in surrounding areas.

Not all ticks require large-bodied wild animals to maintain their populations or to be their source of pathogen infection. For example, *H. elliptica* thrive in peri-urban areas where their preferred hosts, rodents and domestic dogs, coexist. In addition to transovarial transmission of *R. conorii* in ticks, domestic dogs are competent reservoirs of *R. conorii* which can further raise pathogen prevalence in ticks [118]. *Rickettsia conorii* and other members of the SFG *Rickettsia* are overlooked human pathogens that can persist in human dominated landscapes and cause morbidity, mortality, and economic losses in marginalized populations [119]. Comprehensive surveillance of SFG *Rickettsia* in vectors, humans, and animals from endemic areas is needed to inform public health priorities.

## 4. Materials and Methods

### 4.1. Study Area and Sampling Design

This study was conducted in the Lowveld region of the Lubombo province of Eswatini. The region is a flat low-lying savanna on nutrient-rich basaltic soils. The Lowveld experiences mild, dry winters (8–26 °C; 0–50 mm) and hot, wet summers (15–33 °C; 200–500 mm) [120]. The region consists of a variety of land-use types, including government- and privately-owned conservation areas; privately-owned cattle and game ranches; government-owned cattle ranches; commercial agriculture, mainly in the form of intensive sugarcane monocultures; and communal Swazi Nation Land with rural settlements, pasture with livestock, and small-holder croplands.

We sampled questing ticks at 27 sites in the winter of 2018 and resampled 11 sites in the summer 2018 and the winter of 2019 (Figure 6). Sites were in six wildlife conservation areas (Hlane Royal National Park, Mbuluzi Game Reserve, Mlawula Nature Reserve, Dombeya Nature Estate, Mhlosinga Nature Reserve, and Nisela Game Reserve), two mixed cattle and game ranches (Inyoni Yami Swaziland Irrigation Scheme (IYSIS) and Bushlands Ranch), two cattle-only ranches (Nkalashane Ranch and Ndukuyamangendla Ranch), and four communal areas (Lomahasha, Maphiveni, Ndzevane, and Sitsatsaweni). In total, we sampled 21 sites from within protected areas (conservation or mixed ranch) and six sites from within unprotected areas (cattle ranch or communal). See Appendix A for the sampling effort for each site and sampling session. The composition of wildlife in the conservation areas and mixed cattle and game ranches included carnivores (e.g., black-backed jackal, *Canis mesomelas* Schreber, 1775, hyena, *Crocuta crocuta* Erxleben, 1777, and rusty-spotted genet, *Genetta maculata* Gray, 1830), and ungulates (e.g., impala, *Aepyceros melampus* Lichtenstein, 1812, kudu, *Tragelaphus strepsiceros* Pallas, 1766, blue wildebeest, *Connochaetes taurinus* Burchell, 1824, giraffe, *Giraffa camelopardalis* Linneaus, 1758, Southern warthog, *Phacochoerus africanus sundevallii* Lönnberg, 1908, and plains zebra, *Equus quagga burchellii* Gray, 1824) [121,122].

Ticks were sampled by dragging a 1 m^2^ white flannel cloth across the vegetation along standardized transects. The cloth was inspected every 10 m with a maximum of two minutes spent placing ticks directly into 90% molecular-grade ethanol. All ticks (i.e., larvae) remaining on the cloth at the end of a transect were removed using a lint roller. Ticks collected in ethanol were identified morphologically using taxonomic keys [62,92] and stored at −20 °C. All adults and nymphs were identified to species, except for *Rhipicephalus* nymphs. Collected larvae were identified to genus, with the exception of *Rhipicephalus* larvae which were grouped into one-host *Rhipicephlaus* (*Boophilus*) larvae or two- or three-host *Rhipicephalus* larvae based on the presence or absence of festoons [62]. The larvae collected on lint rollers were counted and the total number belonging to each group was calculated from the proportion of larvae identified from the identical site. Molecular techniques were used to verify and augment morphological identification (see Section 4.3).

### 4.2. DNA Extraction

All ticks were individually extracted following an adapted protocol from the Gentra Puregene Tissue Kit (Qiagen). In short, all ticks were rinsed in diH_2_O then 70% ethanol and dried. To process each tick to preserve the exoskeleton for further morphological identification, a cut was made across the posterior-lateral portion of the scutum using a sterile scalpel. Each tick was incubated in cell lysis solution at 65 °C for 15 min followed by the addition of proteinase K and incubated at 55 °C overnight. Then, the proteins were removed, and the DNA was pelleted and washed before suspension in buffer and stored at −20 °C. All adult ticks, all *Amblyomma* nymphs, and up to 20 *Rhipicephalus* nymphs from each site across all sampling session were extracted. A subset of 3 *Amblyomma* larvae, 38 *Rhipicephlaus* (*Boophilus*) larvae, and 46 *Rhipicephalus* larvae were randomly selected across all occurrence sites and extracted for molecular identification.

### 4.3. Molecular Tick Identification

The species-level identification of all adult and nymph *Rhipicephalus* was evaluated using a previously described PCR-RFLP diagnostic assay targeting the *ITS2* gene [63]. The utility of this assay for *Rhipicephalus* species beyond those included in the original study was confirmed in vitro using Geneious Prime^®^ 2021.1.1. After confirmation of variable cut sites for the *Rhipicephalus* species in this study, 25 μL PCR reactions using the *ITS2* primer set and 1 μL of tick DNA at 10–15 ng/μL were used to amplify a 1100 to 1250 bp DNA fragment from specimens (see Appendix A for complete reaction conditions).

Restriction digestion was in a 15 μL volume containing 11 μL PCR-grade water, 1.5 μL 10× buffer (CutSmart^®^, NEB), and 0.5 μL restriction enzyme (0.1 U; *BauI*/*BssSI-v2*; New England Biolabs), along with 2 μL *ITS2* amplicon. The mixture was incubated at 37 °C for 150 min. Digested amplicons were visualized on a 1.5% agarose gel at 140 V for 80 min using RedView™ DNA Gel Stain (GeneCopoeia™). Species identity of each specimen was then assigned based on the digestion profile (Figure 2). Select undigested amplicons were visualized on a 1.5% agarose gel, purified using Exo–Zap, and sent for sequencing (Eurofin Genomics) to confirm PCR-RFLP digestion profiles.

A fragment of the *12S* rDNA sequence and a fragment of the *CO1* sequence was amplified and sequenced for additional molecular characterization of at least two individuals from each tick species. The *12S* sequence was amplified by primers T1B and T2A [64] and 2.5 μL template DNA and the *CO1* sequences was amplified by primers LEP-F1 and LEP-R1 [123] or Chel-*CO1*-F1 and Chel-*CO1*-R1 [124] and 1 μL template DNA (see Table 1 for complete reaction conditions). All conventional PCR reactions were run in a Master Cycler Pro S (Eppendorf, Hamburg, Germany) and included a negative control of PCR-grade water. PCR products were visualized by gel electrophoresis using RedView DNA Gel Stain™ (Genecopoeia™). Positive samples were purified using Exo-SAP (New England Biolabs) and sent for sequencing with the same primers (Eurofin Genomics). Sequences were assembled with Geneious Prime^®^ 2021.1.1.

### 4.4. Pathogen Detection

#### 4.4.1. Spotted Fever Group *Rickettsia*

All adult ticks and *Amblyomma* nymphs were screened by real-time PCR for the presence of bacteria belonging to the genus *Rickettsia*. We used primers and a probe targeting the 17 kd antigen gene to amplify a 114 bp region of *Rickettsia* [125]. Reactions were performed in a 20 μL reaction volume using TaqMan^®^ Universal Master Mix II and 2 μL template DNA (see Appendix A for complete reaction conditions). All positive samples identified by qPCR (CT value of <38) were reevaluated using conventional PCR targeting a 632 bp region of the *ompA* gene [126,127], a 856 bp region of the *ompB* gene [128], and a 770 bp region of the *gltA* gene [129] of spotted-fever group *Rickettsia*. Each reaction used 1 μL template DNA (see Appendix A for complete reaction conditions). A positive control of *Rickettsia amblyommatis* and negative control of PCR–grade water were used in all assays.

#### 4.4.2. Anaplasmataceae

All adult ticks and nymphs were screened by quantitative PCR (qPCR) for the presence of bacteria belonging to *Anaplasmataceae*. We used primers and a probe targeting the *23S* rRNA gene to amplify a 250 bp region of bacteria belonging to the *Anaplasmataceae* family, including the genera *Anaplasma* and *Ehrlichia* [130]. Reactions were performed in a 10 μL reaction volume using QuantiTect^®^ Probe PCR Master Mix and 1 μL template DNA (see Appendix A for complete reaction conditions). All positive samples identified by qPCR (CT value of <38) were reevaluated using conventional PCR targeting a 345 bp region of the 16S rRNA gene that could amplify both *Anaplasma* sp. and *Ehrlichia* sp [131], a set of *Anaplasma* genus-specific primers targeting a 577 bp region of the *rpoB* gene, and a set of *Ehrlichia* genus-specific primers targeting a 609 bp region of the heat shock protein gene (*groEL*) [27]. Each reaction used 1 μL template DNA (see Appendix A for complete reaction conditions). Positive controls included *Anaplasma phagocytophilum* for the *23S* Anaplasmataceae and *rpoB Anaplasma* assays and *Ehrlichia chaffeensis* for the *groEL Ehrlichia* assay. A negative control of PCR-grade water was used in all assays.

#### 4.4.3. Apicomplexa

All adult ticks and nymphs were screened by nested conventional PCR for the presence of protozoans in the phylum Apicomplexa. We used primers targeting the *18S* rRNA gene in all species of piroplasms including the genera *Babesia*, *Theileria*, and *Hepatozoon*. The primary PCR reaction targeted a 487 bp region and used 2 μL template DNA using primers ILO-9029 and ILO-9030. The secondary PCR reaction targeted a 409 bp region and used 2 μL PCR product from the primary reaction using primers MWG4-70 and ILO-7782 (see Appendix A for complete reaction conditions). A positive control of *Theileria cervi* and negative control of PCR-grade water were used in all assays.

### 4.5. Phylogenetic Data Analysis

Alignments were generated for individual gene targets and concatenated genes using MUSCLE implemented in Geneious Prime^®^ 2021.0.3. All alignments were visually inspected prior to phylogenetic inference. Validated sequences obtained from GenBank were included in each alignment. RAxML software [132] was used to estimate maximum likelihood trees using the GTR GAMMA nucleotide model with 1000 bootstrap replicates. Trees were illustrated in R 4.0.5 [133] using treeio [134] and ggTree [135].

*Rickettsia* species were determined by a cycle threshold (CT) value of <38 for the 17 kd real-time PCR screening followed by amplification and sequencing of the *ompA*, *ompB*, and *gltA* genes. We assigned samples to known *Rickettsia* species based on phylogenetic grouping and >98% average sequence similarity across the three genes. Anaplasmataceae species were determined by a CT value of <38 for the *23S* Anaplasmataceae real-time PCR screening followed by amplification and sequencing of the *16S*, *rpoB* (for *Anaplasma* sp.), and/or *groEL* (for *Ehrlichia* sp.). We assigned samples to known *Anaplasma* or *Ehrlichia* species based on phylogenetic grouping and >99% similarity in the *16S* gene region and >98% similarity in the *rpoB* or *groEL* gene regions. We designated the presence of Apicomplexa in these samples based on phylogenetic grouping and >98% sequence identify to the partial *18S* rRNA gene to described species or genotypes.

Samples with undescribed species-level strains were grouped into genotypes based on >98% similarity to one another. For *Rickettsia*, genus-level identity was based on 94–98% sequence similarity to known *Rickettsia*. For *Ehrlichia* or *Anaplasma*, genus-level identity was based on >97% sequence similarity for *16S*, 94–98% for *Ehrlichia groEL*, and <98% for *Anaplasma rpoB* to known *Ehrlichia* and *Anaplasma.* For Apicomplexa, genus-level identity was based on 94–98% sequence similarity to known *Babesia*, *Hepatozoon*, and *Theileria* species. Family-level identification was given to remaining Anaplasmataceae samples with <96% similarity of *16S* to described species. Percent sequence identity cut-offs were determined based on previous phylogenetic studies [136,137,138,139].

### 4.6. Statistical Analyses

To compare the density of questing ticks between protected areas (wildlife conservation areas and mixed cattle and game ranches) and unprotected areas (cattle-only ranches and communal lands), we only used tick counts from the winter of 2018. We aggregated all counts from a site and divided by the total distance sampled to obtain density estimates for each site. Density estimates were log-transformed and compared using two-sample *t*-tests with a 95% confidence interval. To compare the density of questing ticks between years (2018 and 2019) and seasons (winter and summer), we used tick counts from resampled sites. Our density estimates were limited to questing tick species and life stages that could be sampled using tick drags. Statistical comparisons were performed as above, except for using pair *t*-tests.

Pathogen point prevalence and 95% confidence intervals for each tick species and detected pathogen were calculated using the epiR package [140]. The chi-squared test was used to explore associations between tick infection status and site classification (protected vs. unprotected). Tick infection prevalence for each site and sampling session was calculated using the number of infected ticks divided by the total number of ticks tested. All statistical analyses were performed using R 4.0.5 [133].

### 4.7. Voucher Tick and Pathogen Sequences

All pathogen sequences from this study have been deposited in GenBank under the following accession numbers: *Rickettsia* ompA [MZ351038–MZ351047], *Rickettsia* ompB [MZ351048–MZ351055], *Rickettsia* gltA [MZ351056–MZ351064], Apicomplexa 18S [MZ351065–MZ351088], Anaplasmataceae 16S [MZ351089–MZ351099], *Ehrlichia* groEL [MZ351100–MZ351114], and *Anaplasma* rpoB [MZ351115–MZ351122]. All tick sequences from this study have been deposited in GenBank under the following accession numbers: adult *Amblyomma hebraeum* [*12S* = MZ351123, *CO1* = MZ351131, *ITS2* = MZ351142], larval *Amblyomma hebraeum* [*12S* = MZ351124, *CO1* = MZ351132, *ITS2* = MZ35114], adult *Haemaphysalis elliptica* [*12S* = MZ351125, *CO1* = MZ351133, *ITS2* = MZ427481], adult *Rhipicephalus appendiculatus* [*12S* = MZ351126, *CO1* = MZ351134, *ITS2* = MZ351144], larval *Rhipicephalus appendiculatus* [*ITS2* = MZ351143], larval *Rhipicephalus decoloratus* [*CO1* = MZ351135, *ITS2* = MZ351145], adult *Rhipicephalus evertsi* [*12S* = MZ351127, *CO1* = MZ351136, *ITS2* = MZ351146], adult *Rhipicephalus maculatus* [*12S* = MZ351128, *CO1* = MZ351137, *ITS2* = MZ351148], larval *Rhipicephalus maculatus* [*ITS2* = MZ351147], larval *Rhipicephalus microplus* [*CO1* = MZ351138, *ITS2* = MZ351149], adult *Rhipicephalus muehlensi* [*12S* = MZ351129, *CO1* = MZ351139, *ITS2* = MZ351151], larval *Rhipicephalus muehlensi* [*ITS2* = MZ351150], and adult *Rhipicephalus simus* [*12S* = MZ351130, *CO1* = MZ351140, *ITS2* = MZ351152].

## 5. Conclusions

While the description of novel genotypes of *Rickettsia*, *Anaplasma*, *Ehrlichia*, *Theileria*, and other tick-borne pathogens from domestic and wild animals continues to grow [13,14,15,16,22], the accompanying data of associated tick vectors lag behind [23]. In this study, we identified numerous bacterial and protozoan pathogens in host-seeking ticks and made associations between both pathogens and ticks with land-use types composed of distinct communities of vertebrate hosts. Consideration of how human-modified landscapes alter where and how many ticks occur of each along with their speciestick community dynamics will be crucial forin managing the cascading impacts of land conversion on disease risk.

There are still many undescribed species of bacteria and protozoans in ticks with unknown pathogenicity. Surveillance and identification of these potential pathogens are needed to better promote animal and human health in the region. Many ticks have a broader range of associated pathogens than previously known. This study highlights that multiple species of ticks can harbor the same or closely related pathogen species, which could be important in the epidemiology of the diseases and the maintenance of pathogens on the landscape. Looking to the future, disease problems in multi-use landscapes where wildlife and livestock live in proximity are likely to increase and possibly intensify in many regions of southern Africa, as humans continue to expand their footprint and wildlife conservation efforts develop. This changing land use in southern Africa will require ongoing and focused research to understand the determinates of disease at the wildlife–livestock interface, in hope of developing new control strategies that benefit humans, domestic animals, and wildlife.

## Figures and Tables

**Figure 1 pathogens-10-01043-f001:**
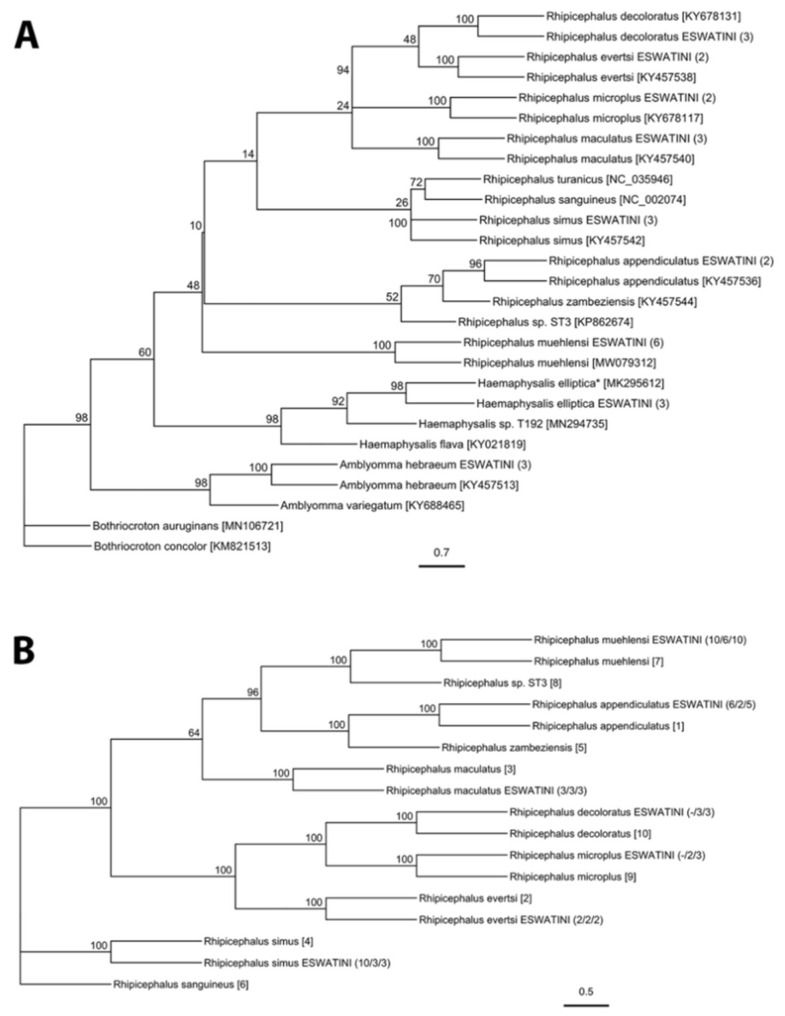
Phylogenetic analysis of tick sequences using (**A**) *CO1* gene for all ticks in study and (**B**) concatenated *12S*, *CO1*, and *ITS2* gene sequences for *Rhipicephalus* species represented as maximum likelihood trees. Bootstrap values at the nodes represent the percent agreement among 1000 replicates. The branch length scale represents substitutions per site. The number of individual specimens that were sequenced from each lineage is indicated in parentheses. For (**A**) the Genbank accession numbers are indicated in brackets and for (**B**) the Genbank accession numbers for *12S*/*CO1*/*ITS2* are: [1] = KY457536/KY457536/KY457500; [2] = KY457538/KY457538/KY457503; [3] = KY457540/KY457540/KY457504; [4] = KY457542/KY457542/KY457508; [5] = KY457544/KY457544/KY457509; [6] = NC_002074/NC_002074/KY945496; [7] = MW080169/MW079312/NA; [8] = NA/KP862674/KP862668; [9] = EU921764/KY678117/KY457506; [10] = KF569940/KY678131/BDU97716. NA = reference sequence not available. * = GenBank sequence unverified.

**Figure 2 pathogens-10-01043-f002:**
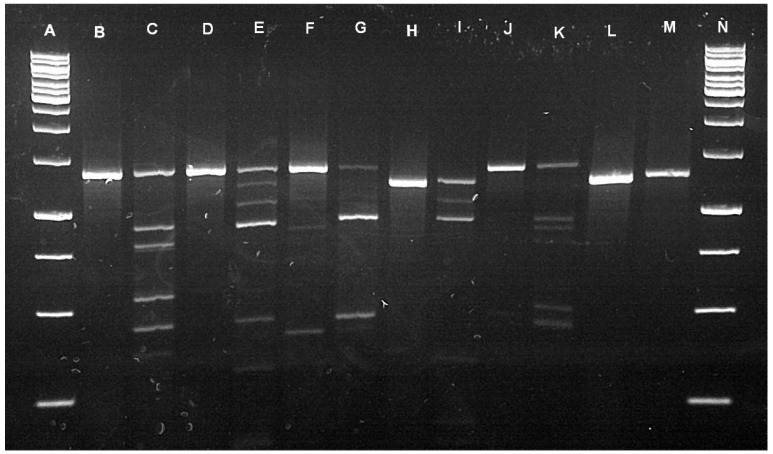
PCR-RFLP digestion profiles. Undigested *ITS2* amplicon and *BauI* restriction digestion profile for variable sequences of the *ITS2* PCR amplicon of *R. appendiculatus* (B = undigested, C = digested), *R. muehlensi* (D = undigested, E = digested), *R. simus* (F = undigested, G = digested), *R. maculatus* (H = undigested, I = digested), *R. microplus* (J = undigested, K = digested), and *R. decoloratus* (L = undigested, M = digested) with 1 kb base pair ladder in A and N.

**Figure 3 pathogens-10-01043-f003:**
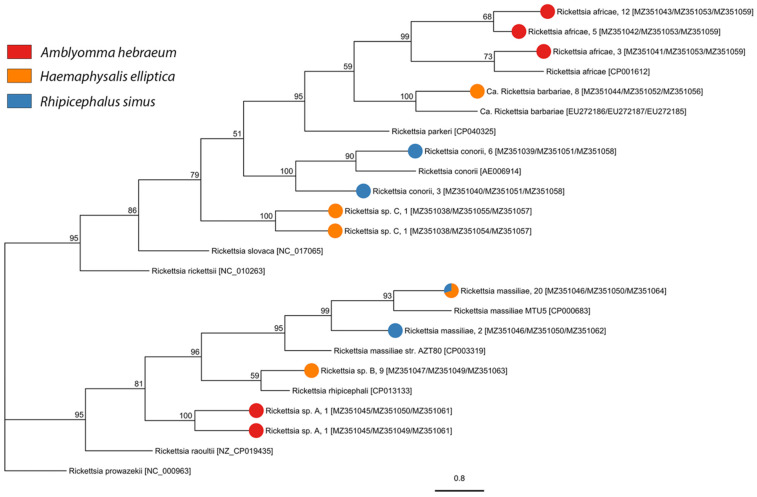
Phylogenetic analysis of *Rickettsia* species and genotypes inferred from concatenated *ompA*, *ompB*, and *gltA* gene sequences using the maximum likelihood method with 1000 bootstraps. The tick species from which each *Rickettsia* isolate was sequenced in this study is represented by the colored pie chart at each tip, followed by the isolate name, the number of samples, and the Genbank accession numbers in square brackets *ompA*/*ompB*/*gltA*). Sequences obtained from GenBank are indicated by their name and accession number. Bootstrap supports, represented as percentages, are indicated on each node. The branch length scale represents substitutions per site.

**Figure 4 pathogens-10-01043-f004:**
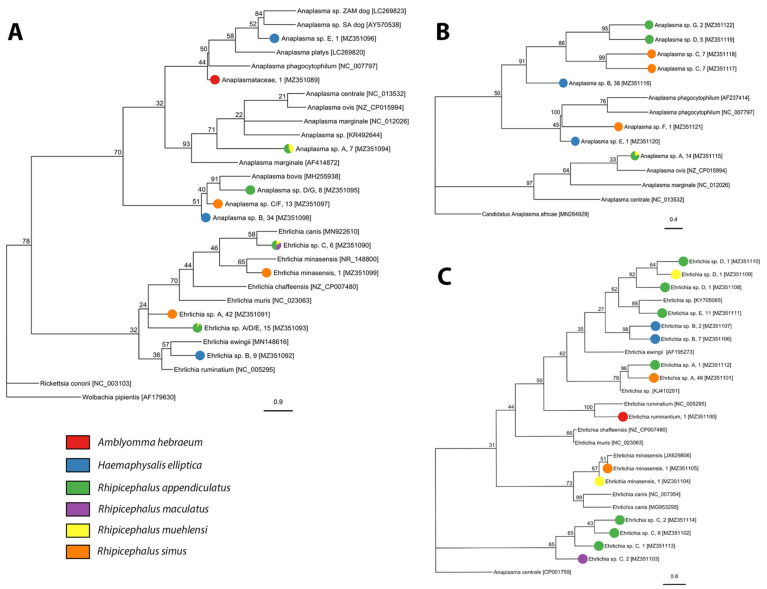
Phylogenetic analysis of Anaplasmataceae species and genotypes found in ticks based on (**A**) *16S*, (**B**) *rpoB*, and (**C**) *groEL* genes using the maximum likelihood method with 1000 bootstraps. The tick species from which of each *Anaplasma* or *Ehrlichia* isolate was sequenced is represented by the colored pie chart at each tip, followed by the isolate name, the number of samples, and the Genbank accession number in square brackets. Sequences obtained from GenBank are indicated by their name and accession number. Bootstrap supports, represented as percentages, are indicated on each node. The branch length scale represents substitutions per site.

**Figure 5 pathogens-10-01043-f005:**
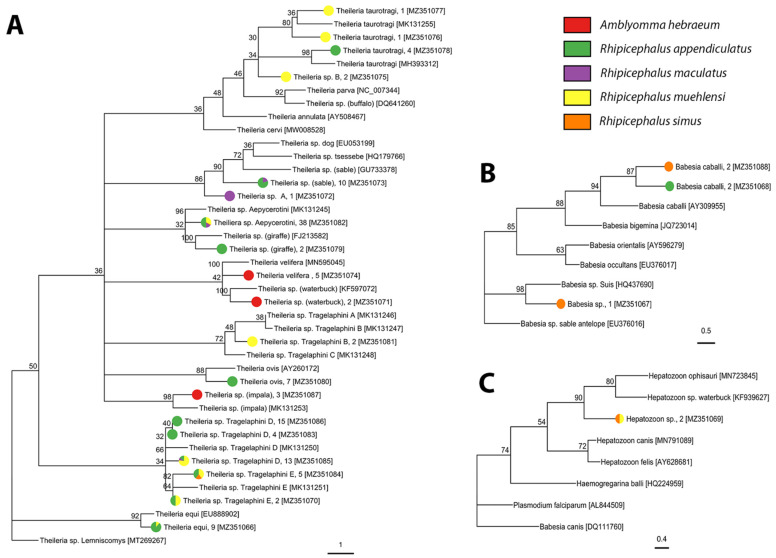
Phylogenetic analysis of (**A**) *Theileria*, (**B**) *Babesia*, and (**C**) *Hepatozoon* species and genotypes found in ticks based on a partial *18S* gene using the maximum likelihood method with 1000 bootstraps. The tick species from which each isolate was sequenced is represented by the colored pie chart at each tip, followed by the isolate name, the number of samples, and the Genbank accession number in square brackets. Sequences obtained from Genbank are indicated by their name and followed by their accession number. Bootstrap supports, represented as percentages, are indicated on each node. The branch length scale represents substitutions per site.

**Figure 6 pathogens-10-01043-f006:**
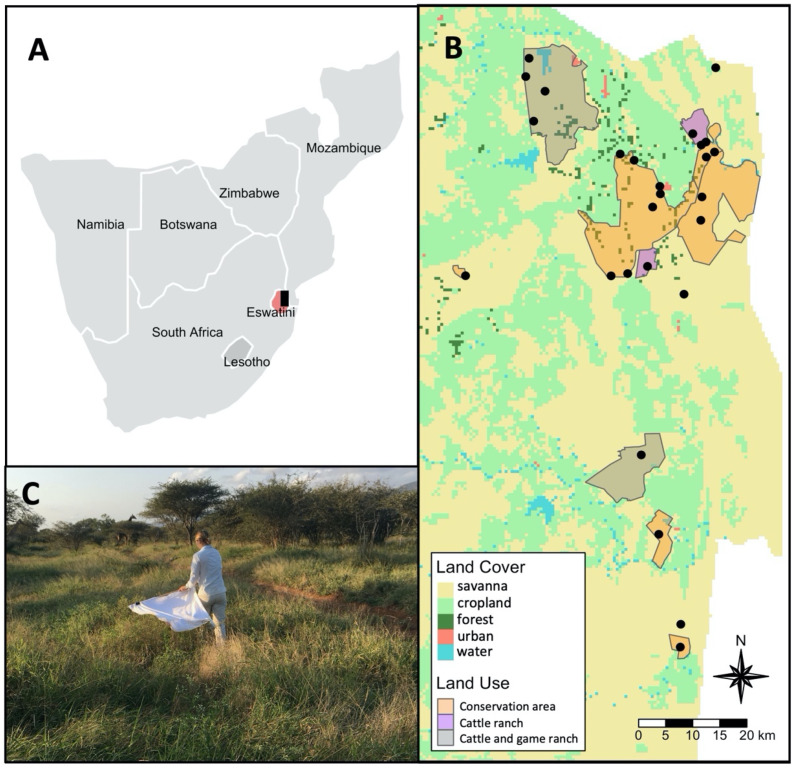
Map of study area. (**A**) Southern Africa showing the location of study area (black box) within Eswatini (red); (**B**) land-use/land-cover map and sampling locations (black dots) within the study area; and (**C**) an example of tick sampling by dragging a white cloth across the vegetation.

**Table 1 pathogens-10-01043-t001:** Sequence homology of the *ITS2*, *CO1*, and *12S* gene regions of ticks in this study with the closest Genbank hit(s). The Genbank accession number is listed after the species names and sequence identity is indicated in brackets. NT = not tested. * = Genbank sequence unverified.

**Tick Species**	***ITS2***	***CO1***	***12S***
*Amblyomma hebraeum*	*Amblyomma hebraeum*: KY457490 (1059/1064 = 99.5%)	*Amblyomma hebraeum*: KY457513 (629/629 = 100%)	*Amblyomma hebraeum*: KY457513 (378/379 = 99.7%)
*Haemaphysalis elliptica*	*Haemaphysalis elliptica*: MK295615 (416/432 = 96.3%) *Haemaphysalis* sp. T192: MN266944 (1042/1049 = 99.3)	*Haemaphysalis ellitptica* *: MK295612 (630/660 = 95.5%) *Haemaphysalis* sp. T192: MN294735 (598/668 = 89.5%)	*Haemaphysalis elliptica*: HM068953 (373/378 = 98.7%)
*Rhipicephalus appendiculatus*	*Rhipicephalus appendiculatus*: KY457500 (1199/1200 = 99.9%)	*Rhipicephalus appendiculatus*: KY457536 (645/645 = 100%)	*Rhipicephalus appendiculatus*: KY457536 (374/276 = 99.5%)
*Rhipicephalus evertsi*	*Rhipicephalus evertsi*: KY457503 (1204/1212 = 99.3%)	*Rhipicephalus evertsi*: KY457538 (702/702 = 99.3%)	*Rhipicephalus evertsi*: KY457538 (384/388 = 99.0%)
*Rhipicephalus maculatus*	*Rhipicephalus maculatus*: KY457505 (1081/1081 = 100%)	*Rhipicephalus maculatus*: KY457540 (629/631 = 99.7%)	*Rhipicephalus maculatus*: KY457540 (381/385 = 99.0%)
*Rhipicephalus muehlensi*	*Rhipicephalus* sp. ST3: KP862668 (1109/1112 = 99.7%) *Rhipicephalus appendiculatus*: KY457500 (1123/1218 = 92.2%)	*Rhipicephalus muehlensi*: MW079312 (496/497 = 99.8%) *Rhipicephalus sp.* ST3: KP862674 (571/645 = 88.5%) *Rhipicephalus appendiculatus*: KY457536 (569/647 = 87.9%)	*Rhipicephalus muehlensi*: MW080169 (317/318 = 99.7%) *Rhipicephalus appendiculatus*: KY457536 (347/377 = 92.0%)
*Rhipicephalus simus*	*Rhipicephalus simus*: KY457508 (1228/1228 = 100%)	*Rhipicephalus simus*: KY457542 (631/634 = 99.5%)	*Rhipicephalus simus*: KY457542 (375/377 = 99.5%)
*Amblyomma* larvae	*Amblyomma hebraeum*: KY457490 (1079/1079 = 100%)	*Amblyomma hebraeum*: KY457513 (629/629 = 100%)	*Amblyomma hebraeum*: KY457513 (378/379 = 99.7%)
*Rhipicephalus* (*Boophilus*) larvae	*Rhipicephalus decoloratus*: MN266919 (1016/1054 = 96.4%)	*Rhipicephalus decoloratus*: KY678131 (634/636 = 99.7%)	NT
	*Rhipicephalus microplus*: KY457506 (1246/1247= 99.9%)	*Rhipicephalus microplus*: KY678117 (635/635 = 100%)	NT
*Rhipicephalus* larvae	*Rhipicephalus appendiculatus*: KY457500 (1159/1160 = 99.9%)	NT	NT
	*Rhipicephalus* sp. ST3: KP862668 (1109/1112 = 99.7%) *Rhipicephalus appendiculatus*: KY457500 (1120/1215 = 92.2%)	NT	NT
	*Rhipicephalus maculatus*: KY457505 (1108/1108 = 100%)	NT	NT

**Table 2 pathogens-10-01043-t002:** SFG *Rickettsia* species identified in Eswatini ticks. Tick species, life stage, and sample size indicated in columns. The numbers of positive samples are shown followed by the prevalence estimates and then the 95% confident intervals for prevalence estimates in brackets. Blue font represents a novel association between a pathogen and tick species. *Rhipicephalus appendiculatus* adults (n = 146), *R. evertsi* adults (n = 6), *R. maculatus* adults (n = 93), and *R. muehlensi* adults (n = 74) were screened for *SFG Rickettsia* and all tested negative.

Pathogen Species/Genotype	*Amblyomma hebraeum*		*Haemaphysalis elliptica*	*Rhipicephlaus simus*
	**Adult (n = 12)**	**Nymph (n = 38)**	**Adult (n = 219)**	**Adult (n = 494)**
***Rickettsia africae***	1; **8.33**% (0.2–38.5%)	19; **50%** (33.4–66.6%)	0	0
**Candidatus *Rickettsia barbariae***	0	0	0	8; **1.6%** (0.7–3.1%)
***Rickettsia conorii***	0	0	9; **4.1%** (1.9–7.7%)	0
***Rickettsia massiliae***	0	0	8; **3.7%** (1.6–7.1%)	14; **2.8%** (1.6–4.7%)
***Rickettsia* sp. A**	0	2; **5.26%** (0.6–17.9%)	0	0
***Rickettsia* sp. B**	0	0	0	9; **1.8%** (0.8–3.4%)
***Rickettsia* sp. C**	0	0	0	2; **0.4%** (0.05–1.5%)
**Richness**	2		2	4

**Table 3 pathogens-10-01043-t003:** Anaplasmataceae identified in Eswatini ticks. Tick species, life stage, and sample size are indicated in columns. The numbers of positive samples are shown followed by the prevalence estimates and then the 95% confident intervals for prevalence estimates in brackets. Blue font represents a novel association between a pathogen and tick species. *Amblyomma hebraeum* adults (n = 12), *R. evertsi* (n = 6), *R. maculatus* nymphs (n = 41), and *R. muehlensi* nymphs (n = 74) were screened for Anaplasmataceae and all tested negative.

	*A. hebraeum*	*H. elliptica*	*R. appendiculatus*		*R. maculatus*	*R. simus*	*R. muehlensi*
Pathogen Species/Genotype	Nymph (n = 38)	Adult (n = 219)	Adult (n = 146)	Nymph (n = 554)	Adult (n = 93)	Adult (n = 494)	Adult (n = 39)
**Anaplasmataceae**	1; **2.6%** (0.1–13.8%)	0	0	0	0	0	0
***Anaplasma*** ** sp. A**	0	0	4; **2.8%** (0.8–6.9%)	7; **1.3%** (0.5–2.6%)	0	0	3; **7.7%** (1.6–20.1%)
***Anaplasma*** ** sp. B**	0	38; **17.4%** (12.6–23.0%)	0	0	0	0	0
***Anaplasma*** ** sp. C**	0	0	0	0	0	13; **2.6%** (1.4–4.5%)	0
***Anaplasma*** ** sp. D**	0	0	3; **2.1%** (0.4–5.9%)	3; **0.5%** (0.1–1.6%)	0	0	0
***Anaplasma*** ** sp. E**	0	1; **0.5%** (0.01–2.5%)	0	0	0	0	0
***Anaplasma*** ** sp. F**	0	0	0	0	0	1; **0.2%** (0.01–1.1%)	0
***Anaplasma*** ** sp. G**	0	0	0	2; **0.4%** (0.04–1.4%)	0	0	0
***Ehrlichia minasensis***	0	0	0	0	0	1; **0.2%** (0.01–1.12%)	1; **2.6%** (0.06–13.5%)
***Ehrlichia ruminantium***	1; **2.6%** (0.1–13.8%)	0	0	0	0	0	0
***Ehrlichia*** ** sp. A**	0	0	2; **1.4%** (0.2–4.9%)	0	0	47; **9.5%** (7.1–12.5%)	0
***Ehrlichia*** ** sp. B**	0	9; **4.2%** (1.9–7.6%)	0	0	0	0	0
***Ehrlichia*** ** sp. C**	0	0	3; **2.1%** (0.4–5.9%)	6; **1.1%;** (0.4–2.3%)	2; **2.2%** (0.3–7.6)	0	0
***Ehrlichia*** ** sp. D**	0	0	1; **0.7%** (0.2–3.8%)	2; **0.4%** (0.04–1.3%)	0	0	1; **2.6%** (0.06–13.5%)
***Ehrlichia*** ** sp. E**	0	0	9; **6.2%** (2.9–11.5%)	2; **0.4%** (0.04–1.3%)	0	0	0
**Richness**	2	3	7		1	4	3

**Table 4 pathogens-10-01043-t004:** Apicomplexa identified in Eswatini ticks. Tick species, life stage, and sample size indicated in columns. Numbers of positive samples are shown followed by the prevalence estimates and then 95% confident intervals for prevalence estimates in brackets. Blue font represents a novel association between a pathogen and tick species. *Haemaphysalis elliptica* adults (n = 219), *R. evertsi* adults (n = 6), and *R. maculatus* adults (n = 93) were screened for Apicomplexa and all tested negative.

	*A. hebraeum*		*R. appendiculatus*		*R. maculatus*	*R. simus*	*R. muehlensi*	
Pathogen Species/Genotype	Adult (n = 12)	Nymph (n = 38)	Adult (n = 146)	Nymph (n = 554)	Nymph (n = 41)	Adult (n = 494)	Adult (n = 39)	Nymph (n = 74)
***Babesia caballi***	0	0	1; **0.7%** (0.02–3.8%)	1; **0.2%** (0.005–1.0%)	0	2; **0.4%** (0.05–1.5%)	0	0
***Babesia*** ** sp.**	0	0	0	0	0	1; **0.2%** (0.01–1.1%)	0	0
***Hepatozoon*** ** sp.**	0	0	0	0	0	1; **0.2%** (0.01–1.1%)	1; **2.6%** (0.1–13.5%)	0
***Theileria equi***	0	0	0	8; **1.4%** (0.6–2.8%)	0	0	1; **2.6%** (0.1–13.5%)	0
***Theileria ovis***	0	0	0	7; **1.2%** (0.5–2.6%)	0	0	0	0
***Theiliera*** ** sp. Aepycerotini**	0	0	0	19; **3.4%** (2.1–5.3%)	6; **14.6%** (5.6–29.2%)	1; **0.2%** (0.01–1.1%)	1; **2.6%** (0.1–13.5%)	11; **14.9%** (7.7–25.0%)
***Theileria*** ** sp. giraffe**	0	0	2; **1.4%** (0.2–4.9%)	0	0	0	0	0
***Theileria*** ** sp. impala**	0	3; **7.9%** (1.7–21.4%)	0	0	0	0	0	0
***Theileria*** ** sp. rodent**	0	0	0	0	0	1; **0.2%** (0.01–1.1%)	0	0
***Theileria*** ** sp. sable**	0	0	7; **4.8%** (1.9–9.6%)	1; **0.2%** (0.005–1.0%)	0	2; **0.4%** (0.05–1.5%)	0	0
***Theileria*** ** sp. Tragelaphini B**	0	0	0	0	0	0	2; **5.1%** (0.6–17.3%)	0
***Theileria*** ** sp. Tragelaphini D**	0	0	8; **5.5%** (2.4–10.5%)	14; **2.5%** (1.4–4.2%)	1; **2.4%** (0.06–12.9%)	0	1; **2.6%** (0.1–13.5%)	9; **12.1%** (5.7–21.8%)
***Theileria*** ** sp. Tragelaphini E**	0	0	1; **0.7%** (0.02–3.8%)	2; **0.4%** (0.04–1.3%)	0	1; **0.2%** (0.01–1.1%)	3; **7.7%** (1.6–20.9%)	0
***Theileria*** ** sp. waterbuck**	0	2; **5.3%** (0.6–17.7%)	0	0	0	0	0	0
***Theileria taurotragi***	0	0	4; **2.7%** (0.8–6.9%)	0	0	0	2; **5.1%** (0.6–17.3%)	0
***Theileria velifera***	1; **8.3%** (0.2–38.5%)	4; **10.5%** (2.9–24.8%)	0	0	0	0	0	0
***Theileria*** ** sp. A**	0	0	0	0	1; **2.4%** (0.06–12.9%)	0	0	0
***Theileria*** ** sp. B**	0	0	0	0	0	0	0	2; **2.7%** (0.3–9.4%)
**Richness**	3		9		3	7	8	

## Data Availability

The data presented in this study are available in the supplementary material and are openly available for viewing within the Open Science Framework at osf.io/z7cve.

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
