# Peer review of "Survey of Ticks and Tick-Borne Rickettsial and Protozoan Pathogens in Eswatini"

_pathogens, 2021, doi:10.3390/pathogens10081043_

Round 1
Reviewer 1 Report
The manuscrispt of Ledger et al., reports the results of a survey of ticks and tick-borne pathogens in Eswatini. The study is well designed and was conducted in a variety of land-use types shared between human rural popolation and animals. In these contexts, a significant number of ticks were collected and analyzed. The methods used are appropriate and a huge amount of data has been obtained.
My main and general comment is to lighten the text and make it more concise and fluent. In particular:
Introduction: I suggest to making the part from lines 51 to 124 more concise
Results (2.2 paragraph): Personally, I had some difficulty reading this part, I suggest to streamlining the text and refer to Table 2 (already exhaustive) for details.
Discussion: I think sub-chapter 3.1 and 3.2, which are closely related to the purpose of the study, should merge and contextualize into a single chapter.
Conclusion: definitely too long, some part and consideration would be more suitable for discussion.
Minor comment:
From line 193 to 198 the authors report some differences between the seasons in the year 2018 and 2019, especially in winter. Do you have found climatic difference between winter 2018 and 2019?
Author Response
Reviewer #1 -----------------------------------------------------------------------------------------------------
The manuscrispt of Ledger et al., reports the results of a survey of ticks and tick-borne pathogens in Eswatini. The study is well designed and was conducted in a variety of land-use types shared between human rural popolation and animals. In these contexts, a significant number of ticks were collected and analyzed. The methods used are appropriate and a huge amount of data has been obtained.
My main and general comment is to lighten the text and make it more concise and fluent. In particular:
Introduction: I suggest to making the part from lines 51 to 124 more concise
We have shortened the introduction in hopes of making the manuscript both more concise and fluent. In particular, we removed background information without direct relevance to the study.
Results (2.2 paragraph): Personally, I had some difficulty reading this part, I suggest to streamlining the text and refer to Table 2 (already exhaustive) for details.
To better convey the results of our study, we removed much of the text from section 2.2 and created a supplementary table (Table S5) to contain relevant details.
Discussion: I think sub-chapter 3.1 and 3.2, which are closely related to the purpose of the study, should merge and contextualize into a single chapter.
We removed the subheading including in the discussion and rearranged topics to fit into a single summary discussion.
Conclusion: definitely too long, some part and consideration would be more suitable for discussion.
Two paragraphs of the conclusion were moved to the discussion section (now lines 428-444) and the remaining conclusion edited for relevance and clarity.
Minor comment:
From line 193 to 198 the authors report some differences between the seasons in the year 2018 and 2019, especially in winter. Do you have found climatic difference between winter 2018 and 2019?
We added a brief remark regarding the climatic differences between the two years of winter sampling and the possible relationship to our findings (lines 413-416).
Reviewer 2 Report
This is a well designed, well written and interesting study. I suggest authors include a statement in line 104 about the mode of transmission of Theileria (transstadial) and Babesia (transovarial), since authors emphasize this factor when discussing rickettsia organisms.
The Introduction section should be shortened.
Author Response
Reviewer #2 -----------------------------------------------------------------------------------------------------
This is a well designed, well written and interesting study. I suggest authors include a statement in line 104 about the mode of transmission of Theileria (transstadial) and Babesia (transovarial), since authors emphasize this factor when discussing rickettsia organisms.
We added a description of the transmission modes for Theileria and Babesia (lines 89-90).
The Introduction section should be shortened.
We have shortened the introduction in hopes of making the manuscript both more concise and fluent
Reviewer 3 Report
This manuscript by Kimberly Ledger and co-authors is a very extensive and detailed survey of ticks and tick-associated microorganisms in the country of Eswatini (prev. Swaziland) in Southern Africa. Rickettsial diseases, particularly Rickettsia africae rickettsiosis, are a significant cause of febrile illnesses and a public health burden in this region of Africa. Anatomic and molecular characterization of hematophagous arthropods native to this region is severely limited. The study design is required extensive sampling but was done responsibly and the data presented provides a significant increase to understanding of ticks and tick-transmitted organisms.
The authors are commended for detailed description and use of statistical analyses, proactive submission and description of DNA sequences, an elegant concluding paragraph (L760-781), and nicely designed figures.
In general, the manuscript is written well (if dense) but a few concerns were noted in the review process. A few concerns/ comments are listed below:
The comparison between protected and unprotected areas does not appear to be particularly relevant as only 2 unprotected sites were assessed. This concern also applies to the discussion/ conclusion where the authors expound upon this finding.
Materials and Methods is not the appropriate place for Fig 6. This is data and should be included in the results or as a supplemental figure. Figure 5 is appropriate for the M&M.
Table 2 is full of wonderful data but is not presented effectively. A four-page table makes appreciation of the data difficult for the reader.
Inclusion of 2 different citation formats is odd and inappropriate. I believe that this also involves the References section.
The discussion needs a sentence or two to describe the fact that the newly identified candidate pathogens may not cause diseases. Many Rickettsiales are considered tick endosymbionts and do produce clinical signs/ symptoms when transmitted to a mammalian host. (This is done appropriately for Thelieria in lines 477-478.)
L152-157: The presence of 39 adult R. muehlensi in a pool of ticks identified as nymphs leads me to believe that the anatomical speciation is poor. Please explain.
Minor Comments:
- Table 3/4 could be moved to the supplement if desired.
- The paragraph from L742-758 lack references.
- Fig 2: I do not understand which known species were chosen to add to this analysis. R. massiliae/ rhipicephali / conorii/ africae/ raoultii seem to be appropriate as these are the closest known relatives to new sequences, but why R. parkeri/slovaka/ rickettsii/ prowazekii?
- Fig 2: Please explain the difference between the 2 Rickettsia sp C, 1 sequences and 2 Rickettsia sp A, 1 sequences.
- Fig 4: Similar statement as above.
- L511: What is T. sp (giraffe)
Author Response
Reviewer #3 -----------------------------------------------------------------------------------------------------
This manuscript by Kimberly Ledger and co-authors is a very extensive and detailed survey of ticks and tick-associated microorganisms in the country of Eswatini (prev. Swaziland) in Southern Africa. Rickettsial diseases, particularly Rickettsia africae rickettsiosis, are a significant cause of febrile illnesses and a public health burden in this region of Africa. Anatomic and molecular characterization of hematophagous arthropods native to this region is severely limited. The study design is required extensive sampling but was done responsibly and the data presented provides a significant increase to understanding of ticks and tick-transmitted organisms.
The authors are commended for detailed description and use of statistical analyses, proactive submission and description of DNA sequences, an elegant concluding paragraph (L760-781), and nicely designed figures.
In general, the manuscript is written well (if dense) but a few concerns were noted in the review process. A few concerns/ comments are listed below:
The comparison between protected and unprotected areas does not appear to be particularly relevant as only 2 unprotected sites were assessed. This concern also applies to the discussion/ conclusion where the authors expound upon this finding.
To clarify the categorization of sites into protected and unprotected, we added a description of the classification to Section 4.1 (study area and sampling design) that includes the number of sites sampled (lines 450-451), in addition to the existing references made on lines 99-102 in Section 1 (introduction) and on line 592-593 Section 4.6 (statistical methods).
Materials and Methods is not the appropriate place for Fig 6. This is data and should be included in the results or as a supplemental figure. Figure 5 is appropriate for the M&M.
Agreed. We have moved the figure to the results section, and it is now labeled as Figure 1.
Table 2 is full of wonderful data but is not presented effectively. A four-page table makes appreciation of the data difficult for the reader.
We agree and have edited the table and for clarity. The information from Table 2 is now presented in Tables 2-4 according to pathogen group. We believe this presentation retains the important data and allows the reader to view an entire table on a single page.
Inclusion of 2 different citation formats is odd and inappropriate. I believe that this also involves the References section.
In this manuscript we have followed the reference format guidelines as outlined by MDPI in their Instructions to Authors for all references of primary literature, books, etc. The discrepancy in citation formatting is a result of including the author citation for scientific names (this refers to the first person or team who makes a scientific name of a taxon available). For this, we followed the formal requirements set under the International Code of Zoological Nomenclature. The code suggests: "The original author and date of a name should be cited at least once in each work dealing with the taxon denoted by that name.” We have done this whenever the author’s name of a family, genus, or species was available. In citing the name of the author, the surname is given in full, (not abbreviated) and followed by the date (true year) of publication in which the name was established is added. The parentheses around the author citation indicate that this was not the original taxonomic placement and used for any scientific name that has been redescribed.
The discussion needs a sentence or two to describe the fact that the newly identified candidate pathogens may not cause diseases. Many Rickettsiales are considered tick endosymbionts and do produce clinical signs/ symptoms when transmitted to a mammalian host. (This is done appropriately for Thelieria in lines 477-478.)
Agreed. We expanded on our original remark of “We also documented three unique genotypes of undescribed Rickettsia with unknown pathogenicity” to include the following sentence in the discussion on lines 306-7: “These Rickettsia genotypes may represent a potential pathogen or a non-pathogenic endosymbiotic rickettsiae.”
L152-157: The presence of 39 adult R. muehlensi in a pool of ticks identified as nymphs leads me to believe that the anatomical speciation is poor. Please explain.
The adult R. muehlensi were misidentified as adult R. appendiculatus, not as nymphs. This fact was unclear as the sentence was originally written. We have clarified the text in lines 123-6 to reflect this.
Minor Comments:
- Table 3/4 could be moved to the supplement if desired.
- Tables 3 and 4 have been moved to supplementary.
- The paragraph from L742-758 lack references.
- Modifications and citations have been added to both paragraphs, now lines 417-422 and lines 629-631.
- Fig 2: I do not understand which known species were chosen to add to this analysis. R. massiliae/ rhipicephali / conorii/ africae/ raoultii seem to be appropriate as these are the closest known relatives to new sequences, but why R. parkeri/slovaka/ rickettsii/ prowazekii?
- Known Rickettsia species included in figure 2 were chosen based on their similarity to sequences obtained during this study and to provide phylogenetic context to less closely related species.
- Fig 2: Please explain the difference between the 2 Rickettsia sp C, 1 sequences and 2 Rickettsiasp A, 1 sequences.
- The two Rickettsia sp A sequences are grouped together based on >98% sequence identity to each other but are named Rickettsia sp. A because they have <98% sequence identity to any described species of Rickettsia. The only differences observed between the two Rickettsia sp A sequences were in the ompB gene.
The same is true for the two Rickettsia sp B sequences.
- Fig 4: Similar statement as above.
- All unique sequences were included in the phylogenetic trees as unique nodes, however species or genotypes names were assigned to sequences based on >98% sequence identity resulting in some nodes on the trees being given the same name and combined during summary statistics.
- L511: What is T. sp (giraffe)
- The name has been changed to include the full genus: Theileria sp (giraffe)